# HICEScore: A Hierarchical Metric for Image Captioning Evaluation

## ABSTRACT

Image captioning evaluation metrics can be divided into two categories, *reference-based* metrics and *reference-free* metrics. However, *reference-based* approaches may struggle to evaluate descriptive captions with abundant visual details produced by advanced multimodal large language models, due to their heavy reliance on limited human-annotated references. In contrast, previous *reference-free* metrics have been proven effective via CLIP cross-modality similarity. Nonetheless, CLIP-based metrics, constrained by their solution of global image-text compatibility, often have a deficiency in detecting local textual hallucinations and are insensitive to small visual objects. Besides, their single-scale designs are unable to provide an interpretable evaluation process such as pinpointing the position of caption mistakes and identifying visual regions that have not been described. To move forward, we propose a novel reference-free metric for image captioning evaluation, dubbed **H**ierarchical **I**mage **C**aptioning **E**valuation **S**core (HICE-S). By detecting local visual regions and textual phrases, HICE-S builds an interpretable hierarchical scoring mechanism, breaking through the barriers of the single-scale structure of existing reference-free metrics. Comprehensive experiments indicate that our proposed metric achieves the SOTA performance on several benchmarks, outperforming existing reference-free metrics like CLIP-S and PAC-S, and reference-based metrics like METEOR and CIDEr. Moreover, several case studies reveal that the assessment process of HICE-S on detailed texts closely resembles interpretable human judgments. The code is available in the supplementary.

## CCS CONCEPTS

• **Computing methodologies** → **Scene understanding**.

## KEYWORDS

Image captioning evaluation, Hierarchical scoring, Hallucination detection, Assessing details in long captions

## 1 INTRODUCTION

Image captioning (IC) [47, 52] is a fundamental task in vision-language (VL) multi-modal learning, which aims to generate descriptions in the natural language given an image. In recent years, IC has attracted more and more attention from researchers since it has wide applications such as helping the visually impaired people [6],

**Unpublished working draft. Not for distribution.**

Permission to make digital or hard copies of all or part of this work for personal or classroom use is granted without fee provided that copies are not made or distributed for profit or commercial advantage and that copies bear this notice and the full citation on the first page. Copyrights for components of this work owned by others than the author(s) must be honored. Abstracting with credit is permitted. To copy otherwise, or republish, to post on servers or to redistribute to lists, requires prior specific permission and/or a fee. Request permissions from permissions@acm.org.

*ACM MM, 2024, Melbourne, Australia*

© 2024 Copyright held by the owner/author(s). Publication rights licensed to ACM.
ACM ISBN 978-x-xxxx-xxxx-x/YY/MM
https://doi.org/10.1145/nnnnnnn.nnnnnnn

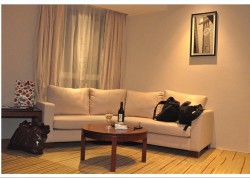

**Brief Caption 1 (BC1):** A living room with a couch and a table.

**Brief Caption 2 (BC2):** A living room with a couch **with pillows** and a table.

| | METEOR | CIDEr | CLIP-S | PAC-S | HICE-S | | | Human |
| | | | | | Global | Local | Overall | |
|---|---|---|---|---|---|---|---|---|
| **BC1** | 43.05 | 350.32 | 0.729 | 0.789 | 0.523 | 0.533 | 0.529 | ✗ |
| **BC2** | 41.82 | 283.32 | 0.728 | 0.807 | 0.542 | 0.536 | 0.538 | ✓ |

**(a) Assessing brief captions.**

**Detailed Caption 1 (DC1):** The image showing a view of a living room. The room is furnished with a white sofa and a wooden coffee table. The walls are adorned with **a patterned wallpaper**, and a window with a curtain is visible. The overall ambiance is warm and inviting.

**Detailed Caption 2 (DC2):** The image showing a view of a living room. The room is furnished with a white sofa and a wooden coffee table. The walls are adorned with **a white decorated painting** and a window with a curtain is visible. The overall ambiance is warm and inviting.

| | METEOR | CIDEr | CLIP-S | PAC-S | HICE-S | | | Human |
| | | | | | Global | Local | Overall | |
|---|---|---|---|---|---|---|---|---|
| **DC1** | 22.39 | 0.00 | 0.720 | 0.778 | 0.515 | 0.537 | 0.526 | ✗ |
| **DC2** | 23.94 | 0.00 | 0.717 | 0.777 | 0.543 | 0.542 | 0.543 | ✓ |

**(b) Assessing detailed captions.**

**Figure 1: Comparisons on evaluation scores of various metrics when assessing (a) brief captions or (b) detailed captions, where correct descriptions about small objects are highlighted in blue and incorrect hallucinations are in golden. The evaluation scores that agree with human judgments are highlighted in green and disagree in red.**

and it also serves as the foundation of other image-to-text tasks like Visual Question Answering [4, 53] and Video Captioning [7, 46]. Based on the encoder-decoder framework, various advanced IC methods have been proposed in the past few years, by extracting meaningful visual information [4, 48, 55], considering effective VL feature alignment [17, 31, 52, 57], and incorporating the knowledge from pre-trained large models [24, 25, 33].

In parallel with the improvement of IC models, increasing efforts have also been dedicated to automatic quality evaluation for generated captions. Besides evaluation, a good metric is also a good learning guidance that can in turn improve the performance of IC models [15, 50].

Traditional referenced-based metrics like BLEU [32], METEOR [5], CIDEr [45], *etc.*, are based on *n-gram* similarity between the generated caption and the human-written references, which focuses on exact phrases matching. However, despite being the most popular measurements for conventional image captioning models, these metrics still have two major limitations that hinder their potential

applications in real-world scenarios. First, human-annotated references are time-consuming and only contain partial visual content. These metrics struggle to correctly identify the visual concepts not appearing in the limited references. As shown in Fig. 1(a), METEOR and CIDEr prefer the brief caption 1 due to "pillows" exist in the image but are not included in the references. Secondly, recent sophisticated multimodal large language models (MLLMs [29, 65]) are inclined to produce detailed captions describing rich details of visual objects. Meanwhile, these detailed captions often exhibit unique linguistic styles that differ from the references. Consequently, traditional metrics often experience a significant decline in performance when assessing those detailed captions due to the notable disparities in sentence length and word choice compared to the references.

To break through the above limitations, with the help of *Contrastive Language-Vision Pretraining* (CLIP) model [33], reference-free metrics CLIP-S [12] and PAC-S [36] are proposed by evaluating the similarity between the given image and the candidate caption using CLIP. Specifically, CLIP projects the original images and sentences to a shared embedding space to calculate cosine similarity between two modalities. Though free of time-consuming human-annotated references and robust in both diverse descriptive styles and sentence lengths, it is often observed that CLIP-S and PAC-S fail to detect hallucinations in the caption, and small objects in the image, as shown in Fig. 1(b). This might be due to the fact that CLIP is trained on web-scale noisy image-text pairs [37], and learns global representations for the whole image and text, ignoring local cross-modality relations among image regions and text phrases [62]. Besides, this inherent characteristic renders them incapable of performing human-like interpretable evaluation capabilities to pinpoint concrete textual mistakes or identify semantic visual regions that may have been omitted.

In general, human experts typically evaluate the quality of image descriptions from two perspectives: *i) Correctness.* Are there any mistakes existing in the captions? If so, where do they occur? *ii) Completeness.* Has all visual content been well comprehensively described? What has been included and what has been left out?

Building on these analyses, in this paper, we propose a novel reference-free metric for IC which is closer to human judgment, named **H**ierarchical **I**mage **C**aptioning **E**valuation **S**core (HICEScore, abbreviated to **HICE-S**) as shown in Fig. 2. Specifically, HICE-S is calculated by gathering the image-caption similarities globally and locally as follows. **For global evaluation**, we follow previous reference-free metrics and employ a pre-trained VL model to extract image and text representations and then compute global image-text similarity based on distances in the shared embedding space. **For local evaluation**, we first construct semantic image regions set and informative text phrases set to decompose the visual content and candidate captions, respectively. Subsequently, these two sets are used to measure local similarity by computing the description *completeness* of semantic regions (recall) and the *correctness* of text phrases (precision). Finally, HICE-S combines the global and local similarities as a fusion score for reference-free IC evaluation. Similar to previous reference-free metrics, HICE-S can also be easily extended to RefHICE-S when human references are provided. Except for image-text compatibility (ITC), RefHICE-S additionally computes hierarchical text-text compatibility (TTC) between references and captions. To evaluate the effectiveness of our proposed metric, we conduct extensive experiments on a series of benchmarks from different perspectives, including the correlation with human judgment, caption pairwise ranking, hallucination sensitivity, and system-level correlation.

Our main contributions can be summarized as follows.

- We proposed a novel hierarchical reference-free metric called HICE-S for IC evaluation. It includes a global score for image-caption compatibility and a local score for region-phrase compatibility.
- Owing to the hierarchical designs, HICE-S provides an interpretable evaluation process that can pinpoint incorrect textual mistakes and present unmentioned visual regions.
- Extensive experiments on typical datasets demonstrate HICE-S achieves SOTA performance on diverse evaluation benchmarks including correlations with human judgments, caption pairwise ranking, and hallucination detection.

## 2 RELATED WORK

Image captioning evaluation methods can be roughly divided into two categories, reference-based and reference-free, according to whether a human-provided reference is available.

### 2.1 Reference-based metric

As references are provided, such as the COCO dataset [28] containing five human-written annotations for every image, we can evaluate the IC performances by calculating the reference-caption similarity, *i.e.*, *text-text compatibility* (TTC). To this end, several *n-gram* matching-based metrics are proposed. Specifically, the initial *n-gram*-based metrics are borrowed from language generation tasks, such as BLEU [32] and METEOR [5] from machine translation; ROUGE [27] from document summarization. Further considering the unique property of IC task that different words in the caption have different importance for describing the given image (*e.g.*, verbs and nouns are always more important than articles), CIDEr [45] and CIDEr-D [45] use TF-IDF to assign weights on different *n-grams*. To overcome the limitations of these methods sensitive to *n-gram* overlap, which is neither necessary nor sufficient for the task of simulating human judgment, SPICE [3] and SoftSPICE [26] are proposed by considering the semantic propositional content defined over scene graphs.

Recently, with the advance of pre-trained large models, some researchers [9, 16, 18, 20, 22, 23, 54] proposed to conduct assessment in the text-feature domain rather than the original word space. For instance, BERT-S [23] and its enhanced version BERT-S++ [54] use the pre-trained BERT [18] to transform word tokens into an embedding space for similarity calculation. Besides, ViLBERTScore [23] and TIGEr [16] are proposed to incorporate visual information into TTC calculation. However, as shown in Fig.1, the above reference-based metrics have a performance degradation when assessing detailed captions, due to the large gap in sentence lengths and visual granularity compared to the brief references.

### 2.2 Reference-free metric

The aforementioned reference-based methods inevitably face that acquiring human-written references for various real-world applications can be prohibitively expensive [36, 41, 51, 64], and it is often

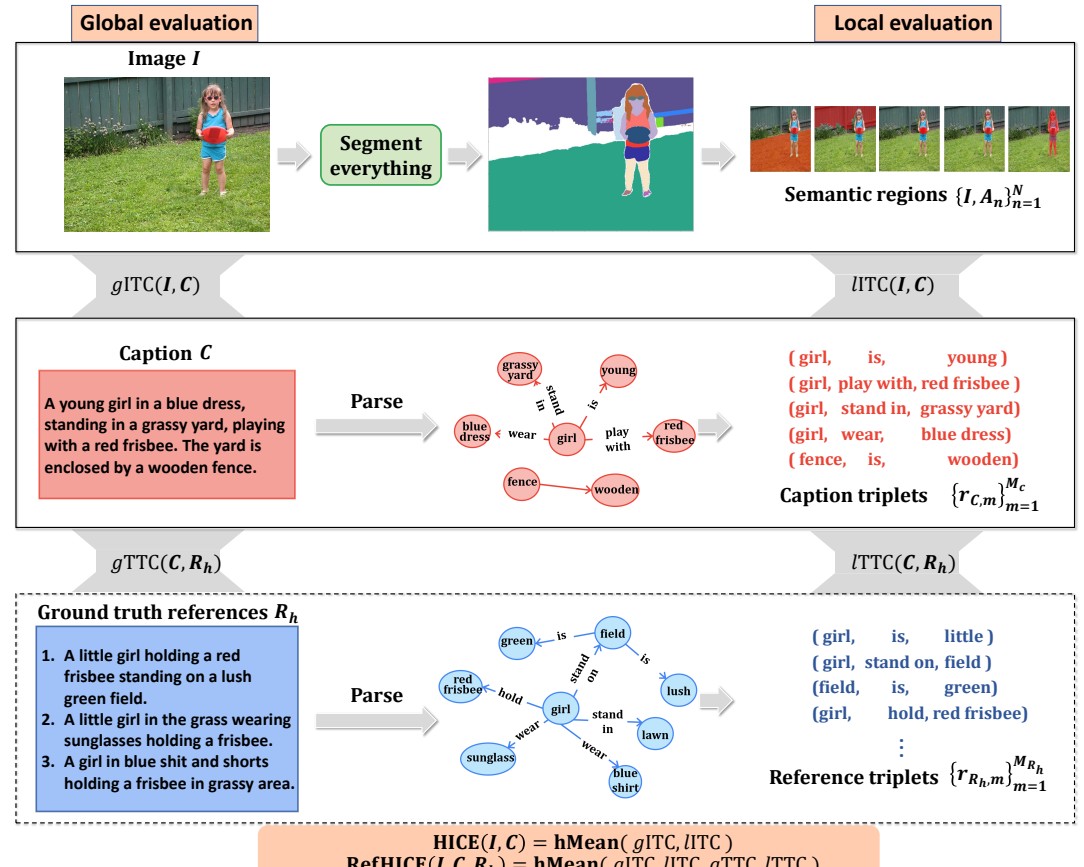

**Figure 2: An illustration of our proposed HICEScore. Left: global image-caption compatibility** gITC **and reference-caption compatibility** gTTC**. Right: local image-caption compatibility** lITC **and reference-caption compatibility** lTTC

.

insufficient to compare generated captions with even multiple references for each image [12]. As a result, there is a large gap remaining between reference-based metrics and human judgments [10, 19]. To solve this problem, reference-free metrics are proposed based on a large VL model CLIP [33], which is trained through the visual-language contrastive loss on a large web-scale image-text-paired dataset [37]. Specifically, CLIP-S [12] is the first reference-free IC metric by using pre-trained CLIP to calculate the image-caption similarity, *i.e.* , *image-text compatibility* (ITC). To further enhance CLIP-S, PAC-S [36] proposes to finetune the CLIP with positive-augmented contrastive learning. Since CLIP-S and PAC-S employ the CLIP text encoder to obtain the sentence embeddings for the calculation of TTC, they can be extended to reference-based metrics by fusing the image-caption (ITC) and reference-caption (TTC) similarities, which are named RefCLIP-S and RefPAC-S, respectively. However, considering only the global evaluation which treats the image and the caption as a whole with CLIP model [33], CLIP-S [12] and PAC-S [36] fail to detect hallucination in the caption and they are not sensitive to visual details in the image. To address fine-grained evaluation, the InfoMetIC builds an object-token similarity score. Nevertheless, InfoMetIC is designed for regular brief captions where an object is generally described by a single token. To

better represent local regions from detailed captions, our proposed HICE-S exploits a graph parser to decompose the descriptive texts into more informative local descriptions rather than a single word.

To provide a more human-correlated evaluation metric, HICE-S considers hierarchical image-text compatibility evaluation, showing a superior ability to assess whether the caption expresses full visual details. It enables HICE-S to perform better for IC evaluations.

## 3 HIERARCHICAL IMAGE CAPTIONING EVALUATION

### 3.1 Problem formulation and preliminaries

Given an image $\mathbf{I}$ and a candidate caption $\mathbf{C}$ for evaluation, our proposed reference-free metric HICE-S is developed for calculating a similarity score between them, denoted as HICE($\mathbf{I}, \mathbf{C}$). Following CLIP-S and PAC-S, we also present a reference-based version RefHICE(I, C, $\mathbf{R}_h$) when human references $\mathbf{R}_h$ are available.

**Alpha-CLIP.** Compared with the vanilla CLIP [33] which is trained via global image-text contrastive learning, Alpha-CLIP [42] is a region-aware enhanced CLIP with an auxiliary alpha channel. Specifically, the alpha channel takes a binary region mask as input

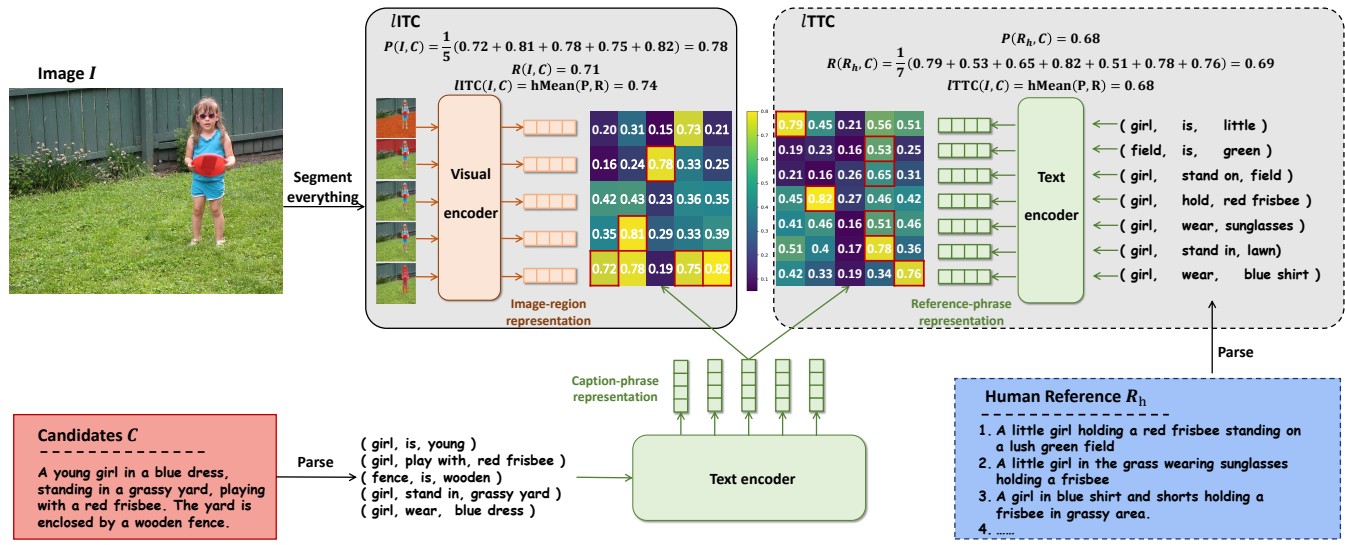

**Figure 3: An illustration of local evaluation including $l$ITC (left part, Eq. 6) and $l$TTC (right part, Eq. 9), where both the precision P and recall R are computed before obtaining the final fusion score through the harmonic mean hMean$(\cdot, \cdot)$.**

to facilitate attention to specific regions. Thus, HICE-S steers Alpha-CLIP to evaluate region-text similarity, as:

$$\text{Alpha} - \text{CLIP}(\mathbf{I}, \mathbf{A}, \mathbf{C}) = \cos\left(C_v(\mathbf{I}, \mathbf{A}), C_t(\mathbf{C})\right), \quad (1)$$

where $\mathbf{A}$ is the binary region mask fed into the alpha channel, $C_v(\cdot), C_t(\cdot)$ are visual encoder and text encoder, respectively. $\cos(\cdot, \cdot)$ denotes the cosine similarity.

## 3.2 Hierarchical evaluation

Although achieving a human-like evaluation effect, the existing reference-free metrics only focus on the global representations of the whole image and text, but ignore the local relations among image regions and text phrases. As a result, they may ignore some visual details such as small objects. To overcome this limitation, as shown in Fig. 2, HICE-S is developed on a hierarchical structure with both global and local similarity evaluations. In practice, the reference-free HICE-S primarily focuses on ITC (image-text: between $\mathbf{I}$ and $\mathbf{C}$), while the reference-based RefHICE-S considers the combination of ITC and TTC (text-text: between $\mathbf{R}_h$ and $\mathbf{C}$), simultaneously.

### 3.2.1 Global evaluation.

As in previous works [12, 36], we transform the image and text to the shared embedding space to compute similarity.

*i) Global ITC*: We use Alpha-CLIP visual encoder $C_v(\cdot)$ and text encoder $C_t(\cdot)$ to extract the features of image $\mathbf{I}$ and caption $\mathbf{C}$, respectively. For feature extraction of the entire image, we utilize a full-one mask $\tilde{\mathbf{A}}$. Then the global ITC can be calculated as

$$\text{gITC}(\mathbf{I}, \mathbf{C}) = \cos\left(C_v(\mathbf{I}, \tilde{\mathbf{A}}), C_t(\mathbf{C})\right). \quad (2)$$

*ii) Global TTC*: We use Alpha-CLIP text encoder $C_t(\cdot)$ to obtain the similarity between human references $\mathbf{R}_h$ and the caption $\mathbf{C}$ as

$$\text{gTTC}(\mathbf{R}_h, \mathbf{C}) = \cos\left(C_t(\mathbf{R}_h), C_t(\mathbf{C})\right). \quad (3)$$

### 3.2.2 Local evaluation.

For local evaluation, we firstly need to obtain local representations for images and texts, and then calculate their similarities. The illustration is shown in Fig. 3.

**Local representations.** We first utilize a pre-trained lightweight segmentation model and a textual graph parser to extract semantic visual regions and informative textual phrases. Then we feed them into the corresponding feature extractor $C_v(\cdot)$ and $C_t(\cdot)$ to obtain the local representations.

*i) Image-region representation set $\mathcal{F}_I$*: Recently, segment-anything models (SAM) [21, 58, 59] has shown powerful capabilities in zero-shot image segmentation field. Many recent works applied SAM to various image-related tasks like medical image analysis [30, 40, 61, 63], object detection [2, 11, 43], and 3D reconstruction [39]. In our work, we use its segment-everything mode to transform the original image to multiple semantic regions with corresponding binary region masks $\{\mathbf{A}_n\}_{n=1}^N$, where $N$ denotes the number of region masks. Then, the Alpha-CLIP visual encoder is employed to obtain the representation for image regions as $f_{I,n} = C_v(I, \mathbf{A}_n)$, which can form a set of image-region representation as $\mathcal{F}_I = \{f_{I,n}\}_{n=1}^N$.

*ii) Text-phrase representation set $\mathcal{F}_C$ for captions $\mathbf{C}$; $\mathcal{F}_{R_h}$ for human references $\mathbf{R}_h$*: Transforming the complete sentences to the text graph whose nodes are nouns while relations are verbs and preposition is effect for language understanding [26]. Motivated by this, given $\mathbf{C}$ (or $\mathbf{R}_h$), we use a textual scene graph parser to get a text graph having $M_c$ (or $M_h$) subject-predicate-object triplets $\{r_{C,m}\}_{m=1}^{M_c}$ (or $\{r_{R_h,m}\}_{m=1}^{M_h}$). Each triplet is equivalent to a short sentence, which is fed into the Alpha-CLIP text encoder to get the representation for these triplets as $f_{C,m} = C_t(r_{C,m})$; $f_{R_h,m} = C_t(r_{R_h,m})$. Then, we obtain the sets of text-phrase representations for caption $\mathbf{C}$ as $\mathcal{F}_C = \{f_{C,m}\}_{m=1}^{M_c}$. for human references $\mathbf{R}_h$ as $\mathcal{F}_{R_h} = \{f_{R_h,m}\}_{m=1}^{M_h}$.

 

| | Flickr8k-Expert [13] | | Flickr8k-CF [13] | | Composite [1] | |
|---|---|---|---|---|---|---|
| | Kendall $\tau_b$ | Kendall $\tau_c$ | Kendall $\tau_b$ | Kendall $\tau_c$ | Kendall $\tau_b$ | Kendall $\tau_c$ |
| BLEU-1 [32] | 32.2 | 32.3 | 17.9 | 9.3 | 29.0 | 31.3 |
| BLEU-4 [32] | 30.6 | 30.8 | 16.9 | 8.7 | 28.3 | 30.6 |
| ROUGE [27] | 32.1 | 32.3 | 19.9 | 8.7 | 30.0 | 32.4 |
| METEOR [5] | 41.5 | 41.8 | 22.2 | 11.5 | 36.0 | 38.9 |
| CIDEr [45] | 43.6 | 43.9 | 24.6 | 12.7 | 34.9 | 37.7 |
| SPICE [3] | 51.7 | 44.9 | 24.4 | 12.0 | 38.8 | 40.3 |
| BERT-S [23] | - | 39.2 | 22.8 | - | - | 30.1 |
| LEIC [9] | 46.6 | - | 29.5 | - | - | - |
| BERT-S++ [54] | - | 46.7 | - | - | - | 44.9 |
| UMIC [22] | - | 46.8 | - | - | - | |
| TIGEr [16] | - | 49.3 | - | - | - | 45.4 |
| ViLBERTScore [23] | - | 50.1 | - | - | - | 52.4 |
| SoftSPICE [26] | - | 54.9 | - | - | - | - |
| MID [20] | - | 54.9 | 37.3 | - | - | - |
| InfoMetIC [14] | - | 55.5 | 36.6 | - | - | 59.3 |
| CLIP-S [12] | 51.1 | 51.2 | 34.4 | 17.7 | 49.8 | 53.8 |
| PAC-S [36] | 53.9 | 54.3 | 36.0 | 18.6 | 51.5 | 55.7 |
| **HICE-S** | 55.9 | 56.4 | 37.2 | 19.2 | 53.1 | 57.9 |
| △ | (+2.0) | (+2.1) | (+1.2) | (+0.6) | (+1.5) | (+1.6) |
| RefCLIP-S [12] | 52.6 | 53.0 | 36.4 | 18.8 | 51.2 | 55.4 |
| RefPAC-S [36] | 55.5 | 55.9 | 37.6 | 19.5 | 53.0 | 57.3 |
| **RefHICE-S** | 57.2 | 57.7 | 38.2 | 19.8 | 53.9 | 58.7 |
| △ | (+1.7) | (+1.8) | (+0.6) | (+0.3) | (+0.9) | (+1.4) |

**Table 1: Human judgment correlation scores on Flickr8k-Expert, Flickr8k-CF and Composite.**

**Local-similarity evaluation.** The calculations of similarity between $\mathcal{F}_I$ and $\mathcal{F}_C$ for local ITC and $\mathcal{F}_{R_h}$ and $\mathcal{F}_C$ for local TTC are shown in Fig. 3. Concretely, we use the cosine similarity to get a pair-wise similarity matrix that denotes the similarity between any two elements from two sets. Then, we use the harmonic mean hMean of precision P and recall R to measure the similarity between two sets.

*i) Local ITC*: We calculate the harmonic mean of precision and recall scores between two sets $\mathcal{F}_I$ and $\mathcal{F}_C$ to get the local ITC as

$$P(\mathbf{I}, \mathbf{C}) = \frac{1}{M} \sum_{m=1}^{M} \max_{1 \le n \le N} \cos(f_{I,n}, f_{C,m}) \quad (4)$$

$$R(\mathbf{I}, \mathbf{C}) = \frac{1}{N} \sum_{n=1}^{N} \max_{1 \le m \le M_c} \cos(f_{I,n}, f_{C,m}) \quad (5)$$

$$l\text{ITC}(\mathbf{I}, \mathbf{C}) = \text{hMean}(P(\mathbf{I}, \mathbf{C}), R(\mathbf{I}, \mathbf{C})), \quad (6)$$

where precision $P(\mathbf{I}, \mathbf{C})$ represents the correctness of captions with respect to the images, while recall $R(\mathbf{I}, \mathbf{C})$ represents the completeness of captions which measures whether all semantic regions are well mentioned.

*i) Local TTC*: In a similar way, we derive the local TTC between two sets $\mathcal{F}_{R_h}$ and $\mathcal{F}_C$ through the harmonic mean of the precision

and recall scores, formulated as:

$$P(\mathbf{R}_h, \mathbf{C}) = \frac{1}{M_c} \sum_{m=1}^{M_c} \max_{1 \le i \le M_h} \cos(f_{R_h,i}, f_{C,m}) \quad (7)$$

$$R(\mathbf{R}_h, \mathbf{C}) = \frac{1}{M_h} \sum_{i=1}^{M_h} \max_{1 \le m \le M_c} \cos(f_{R_h,i}, f_{C,m}) \quad (8)$$

$$l\text{TTC}(\mathbf{R}_h, \mathbf{C}) = \text{hMean}(P(\mathbf{R}_h, \mathbf{C}), R(\mathbf{R}_h, \mathbf{C})). \quad (9)$$

### 3.2.3 Reference-free metric: HICE-S.
Finally, after the combination of global and local evaluation from the ITC view, with gITC in (2), lITC in (6), and we introduce our proposed reference-free metric for IC as

$$\text{HICE}(\mathbf{I}, \mathbf{C}) = \text{hMean}(g\text{ITC}, l\text{ITC}), \quad (10)$$

### 3.2.4 Reference-based metric: RefHICE-S.
Similar to RefCLIP-S [12] and RefPAC-S [36] as extensions of CLIP-S and PAC-S, respectively, when the human-provided references $\mathbf{R}_h$ are available, we can present a reference-augmented version of HICE-S, *i.e.* RefHICE-S, which achieves higher correlation with human judgment. Specifically, we just need to further consider the global TTC score, gTTC in Eq. 3 and local TTC score lTTC in Eq. 9. Then, same as Eq.10 to calculate the harmonic mean of those four items, we have the RefHICE − S$(\mathbf{I}, \mathbf{C}, \mathbf{R}_h)$.

$$\text{RefHICE}(\mathbf{I}, \mathbf{C}, \mathbf{R}_h) = \text{hMean}(g\text{ITC}, l\text{ITC}, g\text{TTC}, l\text{TTC}). \quad (11)$$

| | HC | HI | HM | MM | Mean |
|---|---|---|---|---|---|
| length | 51.7 | 52.3 | 63.6 | 49.6 | 54.3 |
| BLEU-1 [32] | 64.6 | 95.2 | 91.2 | 60.2 | 77.9 |
| BLEU-4 [32] | 60.3 | 93.1 | 85.7 | 57.0 | 74.0 |
| ROUGE [27] | 63.9 | 95.0 | 92.3 | 60.9 | 78.0 |
| METEOR [5] | 66.0 | 97.7 | 94.0 | 66.6 | 81.1 |
| CIDEr [45] | 66.5 | 97.9 | 90.7 | 65.2 | 80.1 |
| BERT-S [23] | 65.4 | 96.2 | 93.3 | 61.4 | 79.1 |
| BERT-S++ [54] | 65.4 | 98.1 | 96.4 | 60.3 | 80.1 |
| TIGEr [16] | 56.0 | 99.8 | 92.8 | 74.2 | 80.7 |
| ViLBERTScore [23] | 49.9 | 99.6 | 93.1 | 75.8 | 79.6 |
| FAIEr [49] | 59.7 | **99.9** | 92.7 | 73.4 | 81.4 |
| MID [20] | 67.0 | 99.7 | 97.4 | 76.8 | 85.2 |
| InfoMetIC [14] | 69.9 | 99.7 | 96.8 | 79.6 | 86.5 |
| CLIP-S [12] | 55.9 | 99.3 | 96.5 | 72.0 | 80.9 |
| PAC-S [36] | 60.6 | 99.3 | 96.9 | 72.9 | 82.4 |
| **HICE-S** | 68.6 | 99.7 | 96.5 | 79.5 | 86.1 |
| △ | (+8.6) | (+0.4) | (-0.4) | (+6.6) | (+3.7) |
| RefCLIP-S [12] | 64.9 | 99.5 | 95.5 | 73.3 | 83.3 |
| RefPAC-S [36] | 67.7 | 99.6 | 96.0 | 75.6 | 84.7 |
| **RefHICE-S** | 71.4 | 99.7 | 97.7 | 79.7 | 87.3 |
| △ | (+3.7) | (+0.1) | (+1.7) | (+4.1) | (+2.6) |

**Table 2: Accuracy results on the Pascal-50S dataset [45] obtained by averaging the scores over five random draws of reference captions (except for reference-free metrics). Detailed descriptions about HC, HI, HM and MM can be found in Sec. 4.3.**

## 4 EXPERIMENTS

In this section, we conduct comprehensive experiments including *correlation with human judgments* in Sec. 4.2, *caption pairwise ranking* in Sec. 4.3, *object hallucination sensitivity* in Sec. 4.4, and *system-level correlation* in Sec. 4.5, to demonstrate the superiority of our proposed metrics HICE-S and RefHICE-S. Furthermore, we conduct ablation experiments and present some discussion to investigate the roles of each part in our metric.

### 4.1 Implementation details

Owing to the strong generalization capabilities of large models, HICE-S and RefHICE-S employ distinct pre-trained models to achieve different functionalities without further fine-tuning. We choose Alpha-CLIP-ViT-L/14 to compute global and local similarity for both ITC and TTC. In the local evaluation, we utilize a fast and lightweight segment-everything model mobileSAMv2 [59] to extract semantic image region masks. For the textual phrases extraction, we need to transform a sentence to several subject-predicate-object phrases, which is realized by TextGraphParser [26].

**Compared metrics.** Following previous works [12, 36], we mainly compare our metrics with three types of metrics, one for reference-based and one for reference-free. *i) n-grams-based metric* is the reference-based one including BLEU [32], METEOR [5], ROUGE [27], CIDEr [45], and SPICE [3]. *ii) embedding-based metric* is another type of the reference-based one including BERT-S [23], BERT-S++ [54], LEIC [9], TIGEr [16], UMIC [22], ViLBERTScore [23],

| | FOIL [38] | |
|---|---|---|
| | Acc. (1 Refs) | Acc. (4 Refs) |
| BLEU-1 [32] | 65.7 | 85.4 |
| BLEU-4 [32] | 66.2 | 87.0 |
| ROUGE [27] | 54.6 | 70.4 |
| METEOR [5] | 70.1 | 82.0 |
| CIDEr [45] | 85.7 | 94.1 |
| MID [20] | 90.5 | 90.5 |
| CLIP-S [12] | 87.2 | 87.2 |
| PAC-S [36] | 89.9 | 89.9 |
| **HICE-S** | 93.1 | 93.1 |
| △ | (+3.2) | (+3.2) |
| RefCLIP-S [12] | 91.0 | 92.6 |
| RefPAC-S [36] | 93.7 | 94.9 |
| **RefHICE-S** | **96.4** | **97.0** |
| △ | (+2.7) | (+2.1) |

**Table 3: The accuracy results on the FOIL hallucination detection datasets [38] with one reference and four references, respectively.**

SoftSPICE [26], and MID [20] and infoMetIC [14]. *iii) CLIP-based metric* is recently proposed reference-free one including CLIP-S [12] and PAC-S [36] as well as their reference-based extension RefCLIP-S and RefPAC-S. Our metric HICE-S belongs to the reference-free one and its reference-based extension is RefHICE-S.

### 4.2 Correlation with human judgments

Given an image-caption pair, this task is to see whether the scoring of the metric is consistent with the human scoring (often contains more than one person and takes average). To this end, we conduct the experiment on three widely adopted human judgment datasets Flickr8k-Expert, Flickr8k-CF, and Composite [1, 13]. Following previous approaches [12, 36], we compute both Kendall $\tau_b$ and Kendall $\tau_c$ correlation scores for each dataset.

**Dataset.** The *Flickr8k-Expert* [13] contains three expert human rating scores for each image-caption pair, with 5,664 images and 17k expert annotations in total. The evaluation scores range from 1 to 4 and the higher the better. The *Flickr8k-CF* [13] is collected from CrowdFlower [44] with 48k image-caption pairs. The assessment on Flickr8k-CF is binary where "yes" ("no") represents that the candidate caption is (not) relevant to the image. Each image-caption pair has more than three scores. Following [12, 36], we use the mean proportion of "yes" annotations as the final score for each pair to compute the correlation with human judgments. The *Composite* [1] is composed of 12k human judgments with 3,995 images from Flickr8k [13] (997 images), Flickr30k [56] (991 images), and COCO [28] (2007 images). Each image-caption pair is evaluated with a human score, ranked from 1 to 5 where a higher score indicates the better matching between the caption and the image.

**Results.** Results on Flickr8k-Expert, Flickr8k-CF, and Composite are reported in Table 1. Our proposed metric achieves the best correlation results with human judgment compared to previous metrics. Specifically, on the Flickr8k-Expert, the **HICE-S** and **RefHICE-S** exhibit significant advantages over the SOTA score PAC-S [36] and

| Methods | M | C | CLIP-S | PAC-S | precision | recall | HICE-S |
|---------|---|---|--------|-------|-----------|--------|--------|
| | | | Brief Captions | | | | |
| $\mathcal{M}^2$ Transformer [8] | 28.7 | 127.9 | 0.605 | 0.806 | 0.536 | 0.516 | 0.539 |
| VinVL [60] | 31.1 | 140.9 | 0.627 | 0.821 | 0.540 | 0.545 | 0.553 |
| BLIP-2 [24] | **32.4** | **144.2** | **0.635** | **0.829** | **0.551** | **0.557** | **0.568** |
| | | | Detailed Captions | | | | |
| LLaVA-1.6 [29] | | | | | | | |
| length=10 | 15.3 | 42.6 | 0.588 | 0.746 | **0.518** | 0.516 | 0.522 |
| length=25 | **19.8** | 35.6 | **0.603** | **0.748** | 0.511 | 0.526 | **0.535** |
| length=65 | 14.8 | 1.1 | 0.591 | 0.725 | 0.480 | 0.533 | 0.513 |
| length=75 | 13.1 | 0.0 | 0.590 | 0.729 | 0.478 | **0.536** | 0.515 |

**Figure 4: Different scores of previous SOTA captioning models on COCO testing dataset [28].**

RefPAC-S [36], with more than 1.5 points gained in terms of both $\tau_b$ and $\tau_c$. On the Flickr8k-CF and Composite, our metrics also perform better than previous models, in particular under the reference-free scenarios.

### 4.3 Caption pairwise ranking

Given an image with two captions, this task is to see whether the better caption selected by metric is consistent with the human choice.

**Dataset.** *Pascal-50S* [45] contains 1k images collected from UIUC PASCAL Sentence dataset [34] and 4k sentence pairs, which are split into four groups, two *correct* captions written by *human* (HC); one *incorrect* and one *correct* caption by *human* (HI); two correct captions with one written by *human* and the other one by *machine* (HM); two correct captions generated by *machine* (HM). Each caption pair has 48 human judgments about which one better matches the image. The vote is based on the principle of majority rule, while a random selection will be operated in case of the equal number of votes. Following [36], we compute the accuracy instead of correlation coefficients and select 5 references among the 48 human references to serve as ground truth to compute reference-based metrics (average over 5 random draws of references).

**Results.** In Table. 2, we provide the accuracy results. Obviously, under the reference-free scenarios, our proposed HICE-S outperforms SOTA metrics, CLIP-S[12] and PAC-S [36] by a large margin, about 3.7 points improvements on average accuracy of four categories, as shown in the "Mean" column. Even compared to reference-based scoring methods, our HICE-S metric is still in the lead with **6** points better than CIDEr [45], 2.8 points than RefCLIP-S [12] and 1.4 points higher than RefPAC-S [36]. Such an excellent performance can be attributed to the introduction of hierarchical evaluation designs, which are more in accordance with human criteria. Furthermore, our reference-based metric RefHICE-S also achieves superior accuracy results compared to previous methods.

### 4.4 Object hallucination Sensitivity

As mentioned by Rohrbach in [35], current image captioning models are prone to generate objects that are not present in the images,

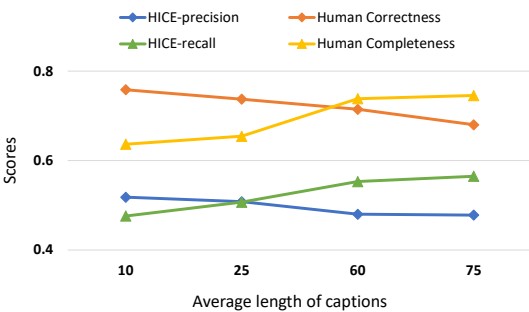

**Figure 5: HICE-S's local ITC precision and recall evaluation scores with different caption lengths compared to human correctness and completeness scores.**

*aka*, object hallucinations. Thus, it is critical to identify object hallucinations for IC evaluation

**Dataset.** *FOIL* dataset [38] contains 32k images from COCO validation set [28]. Each image is accompanied by a correct-foil caption pair, where the correct sentence is collected from the original COCO annotations [38] while the foil sentence is generated by replacing a noun with a similar but incorrect description. Following previous works [12, 36], the accuracy is derived as the percentage of higher scores assigned to the correct caption

**Results.** The results are available in Table 3 under both one reference-available case and four reference-available case [12, 36]. Note that since the reference-free methods do not use reference, the performance is the same between the two columns. It can be observed that our proposed HICE-S and RefHICE-S outperform the existing best metric PAC-S [36] with a large margin, especially for reference-free evaluation. For reference-based cases, the improvement of RefHICE-S on "1 Refs" is higher, which is more difficult than "4 Refs", since reference is useful to find hallucinations [12, 36].

### 4.5 System-level correlation

We also propose to evaluate the efficaciousness of HICE-S and RefHICE-S within image captioning models. Table 4 has shown the evaluation results on images from the COCO testing set [28] for both brief caption generations and detailed caption generations. The results about different evaluation metrics are exhibited, including BLEU-4 [32] (B@4), METEOR [5] (M), CIDEr [45] (C), CLIP-S [12], PAC-S [36], and our HICE-S.

For conventional brief caption evaluation, we assess the caption generations of some SOTA methods including $\mathcal{M}^2$ Transformer [8], VinVL [60], and BLIP-2 [24]. As we can see, both reference-based metrics (METEOR, CIDEr) and reference-free metrics (CLIP-S, PAC-S, and our proposed HICE-S) can effectively evaluate the previous image captioning methods and identify the best captioning model.

For detailed caption evaluation, we leverage an advanced multi-modal large language model LLaVA-1.6 [29] to generate detailed captions of different caption lengths with a prompt template "*describe the image scene in {length} words*". As we can see, CIDEr is unable to assess captions whose lengths are larger than 65 words, presenting nearly zero scores. HICE-S in Table. 4 has shown that the

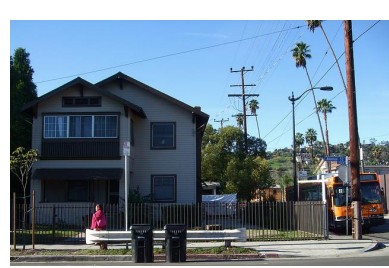

The image showing a different view of a residential area. The photos capture the a house, which has a distinctive black roof and a white wall. The house is surrounded by palm trees and a fence, and there are benches and a truck visible in the background. A yellow dog is running on the sidewalk.

**Correct > 0.5**
— house have roof 0.537
— trees surround house 0.542
— fence surround house 0.563
— benches 0.521
— truck in the background 0.530
— house have white wall 0.590

**Incorrect < 0.5**
— yellow dog 0.312
— dog run on sidewalk 0.369
— residential area 0.474

**Mentioned > 0.5**

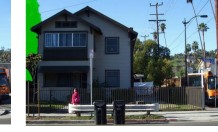
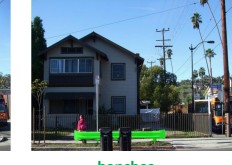
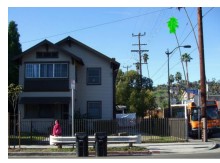

fence surround house 0.563 | trees surround house 0.512 | benches 0.521 | palm trees surround house 0.545

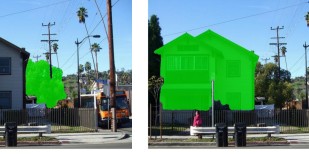

trees surround house 0.502 | truck in the background 0.530 | trees surround house 0.542 | house have white wall 0.590

**Unmentioned < 0.5**

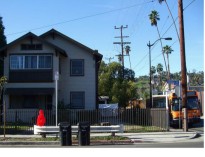
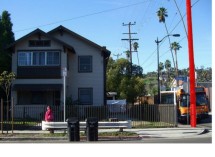
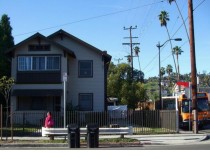
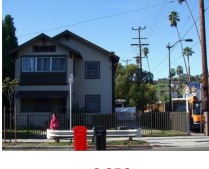

0.390 | 0.473 | 0.376 | 0.356

**Figure 6: A Local evaluation example of HICE-S.** *Left top* is the original image and detailed caption. *Left bottom* are the local text phrases and corresponding precision scores that measure the correctness of each phrase. Text phrases with scores lower than 0.5 are considered as incorrect phrases and larger than 0.5 as correct . *Right* are the semantic regions and corresponding recall scores that measure the completeness of the captions. Those semantic regions with scores lower than 0.5 are considered as unmentioned. For the mentioned regions, we have presented the matched text phrase with the highest local similarity.

| | | Flickr8k-Expert | | Pascal-50S | FOIL |
|---|---|---|---|---|---|
| global | local | Kendall $\tau_b$ | Kendall $\tau_c$ | Acc. | Acc.(1 Ref) |
| ✓ | ✗ | 52.3 | 53.1 | 82.1 | 90.7 |
| ✗ | ✓ | 54.6 | 54.4 | 84.8 | 92.0 |
| ✓ | ✓ | **55.9** | **56.4** | **86.1** | **93.1** |

**Table 4: Ablation studies of our proposed HICE-S on Flickr8k-Expert dataset [13], Pascal-50S [45] and FOIL dataset [38]. global and local denote global and local evaluation, respectively.**

longer the caption, the lower the precision score, and conversely, the higher the recall score. This demonstrates that although longer captions can express more visual content, they often tend to introduce more errors and hallucinations. To further explore the correlation with human judgments, we invite 5 human experts to assess the LLaVA captions from both correctness and completeness perspectives. Fig.5 has shown the comparison results between human assessment and HICE local assessment. The high correlation demonstrates the effectiveness of our proposed hierarchical evaluation.

## 4.6 Ablation study and discussion

To further investigate the roles of each part in our proposed metric, we conduct the ablation experiments on Flickr8k-Expert [13], Pascal-50S [45] and FOIL [38].

**Hierarchical evaluation.** The first row of Table 4 displays the performance results of the individual global evaluation, while the second row shows the individual local evaluation results. The third row presents the combined results of global and local evaluations. When adopting a hierarchical evaluation, there is significant performance improvements on Flickr8k-Expert, Pascal-50S, and FOIL. Finally, our HICE-S achieves the SOTA results on all datasets by hierarchical evaluation.

**Interpretable evaluation process.** As presented in Fig. 6, HICE-S can perform human-like interpretable evaluation process. For correctness evaluation, HICE-S can effectively recognize correct phrases and incorrect phrases that are not included in the image content. For completeness evaluation, HICE-S can also identify the mentioned region and match textual phrases.

## 5 CONCLUSION

In this paper, we propose a novel reference-free metric for image captioning evaluation, dubbed HICE-S. We introduce a hierarchical evaluation design, which includes the global evaluation, built on the image-level and the sentence-level representations, and the local evaluation, based on the region-level and the phrase-level features. The combination of global evaluation and local evaluation renders our HICE-S more sensitive to object hallucination and minor errors. Extensive experiments demonstrate that our proposed HICE-S and the reference-based RefHICE-S are superior to all previous metrics including reference-free and reference-based methods on correlations with human judgments, caption pairwise ranking, and hallucination detection.

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
