# OpenReview forum: "HICEScore: A Hierarchical Metric for Image Captioning Evaluation"
_acmmm.org/ACMMM/2024/Conference — MM2024 Poster_

### Official Review · Reviewer_pmAA · 2024-05-20

**Rating:** 4
**Confidence:** 4

**Summary:**

The paper introduces a hierarchical evaluation metric for image captioning, called Hierarchical Image Captioning Evaluation Score (HICE-S), that focuses on capturing more detailed descriptions. This novel approach aims to better align with human judgments and overcome the limitations of existing metrics that either rely too heavily on limited human-annotated references or fail to detect local textual hallucinations and small visual objects. Specifically, the authors develop a global and local similarity evaluation to better align with human judgments.  They demonstrate the effectiveness of the HICE metric across several datasets.

**Strengths:**

* The claim is clear and also well supported through a range of experiments.
* The motivations are strong, grounded in real-world problems, and are driven by some problems observed in the currently used metrics.
* The ability of the model reported in Figure 6 is extremely interesting

**Limitations:**

* The paper uses the Alpha-CLIP-ViT-L/14 backbone, while some competitors, such as CLIPScore and PACScore, employ the CLIP ViT-B/32 backbone.  This discrepancy makes the comparison reported in the tables unfair.
It would be more fair to report in all the main tables the Clip ViT-L/14 or OpenClip ViT-L/14 CLIP-S and PAC-S scores.
* The model's use of a detector to obtain masks increases computational requirements as the number of objects present in the image grows.  It would be interesting to compare the time required to compute the metric when varying the number of detected masks.
Moreover, for all the evaluation datasets (Flickr-Expert, CF, etc..) the proposed model requires the computation of masks to feed as input to the Alpha-CLIP backbone. The runtime required for the entire pipeline should be compared with the time needed to compute scores for other metrics
* Clarification is needed on whether both the Alpha-CLIP and mobileSam models are frozen.
* Minor: Figure 3 should be more readable. It is suggested to reduce the image I size to emphasize the segmentation mask and lighten the formula part for better clarity.

**Suitability:**

3

---

### Official Review · Reviewer_Xkh4 · 2024-05-26

**Rating:** 2
**Confidence:** 4

**Summary:**

The paper introduces HICE-S, a reference-free metric for evaluating image captioning. The proposed of HICE-S aims to overcome the limitation of reference-based and CLIP-based metrics methods. It uses a hierarchical approach to assess both global image-text compatibility and local region-phrase matches, allowing for more detailed and accurate evaluations. HICE-S is shown to be highly effective in benchmarks, outperforming other metrics in correlating with human judgment and detecting inaccuracies in captions.

**Strengths:**

1. The paper is well-written and easy to follow.
2. This paper proposes a local evaluation strategy to better model the similarity between visual concepts and phrases with the help of segment anything. It also incorporates a method to aggregate hierarchical scores.
3. The author conducts extensive experiments across multiple datasets, which suggest the data generalizability of the proposed method.

**Limitations:**

1. The contribution seems to be limited. The previous reference-free evaluations (e.g., InfoMetIC [1]) also consider the fine-grained region-word level similariy. And in this paper, the author uses a mask-phrase level similariy with the help of  segment anything. Are the performance improvment taken from the stronger detector (sam vs. bottom-up detector)?
2. The author state InfoMetIC is a reference-based method (L635-L662), however InfoMetIC is a reference-free method as stated in its paper.
3. The experiment evaluate recent MLLMs such as LLAVA-1.6 and BLIP-2. It may be beneficial to include results for GPT4V to enhance the reliability of the experimental findings.
4. The HICE-S rely on segment anything, which visual encoder is also widely used by some recent proposed MLLMs (e.g., DeepSeek-VL [2]). Does this reliance on segment anything make HICE-S evaluations potentially unfair towards these models?
[1] Hu, A., Chen, S., Zhang, L., & Jin, Q. (2023). InfoMetIC: An Informative Metric for Reference-free Image Caption Evaluation. arXiv preprint arXiv:2305.06002.
[2] Lu, H., Liu, W., Zhang, B., Wang, B., Dong, K., Liu, B., ... & Ruan, C. (2024). DeepSeek-VL: towards real-world vision-language understanding. arXiv preprint arXiv:2403.05525.

**Suitability:**

3

---

### Official Review · Reviewer_Y7H2 · 2024-05-26

**Rating:** 4
**Confidence:** 4

**Summary:**

For developing reference-free metrics for evaluating the quality of detailed image captioning, this work proposes a metric namely HICEScore by leveraging a segmentation model to detect visual region masks, a text model to extract graphs, and a vision-language model to calculate the cross-modal similarity. Without extra training, their metric achieves state-of-the-art correlation with human judgment on multiple benchmarks.

**Strengths:**

1. A reference-free metric achieves SOTA correlation with human judgment without extra training.
2. They perform sufficient experiments on multiple benchmarks to verify the effectiveness of their model.

**Limitations:**

1. InfoMetIC is a reference-free metric while wrongly classified as reference-based metrics in lines 635-662. Therefore, HICE-S doesn't achieve SOTA performance on Pascal-50S and Composite under reference-free scenarios, descriptions in lines 716-718 and 738-741. However, I still recognize the effectiveness of HICE-S because InfoMetIC should be fine-tuned on captioning datasets while HICE-S is not needed. Authors should give more discussion about the comparison with InfoMetIC about whether need extra training during experiments.

2. Both InfoMetIC and HICE-S consider global and local similarity. Since HICE-S is declared more suited for evaluating detailed descriptions, a direct correlation comparison of InfoMetIC and HICE-S with human judgment on detailed descriptions should be given. For example, authors can add extra two lines of InfoMetIC to Figure 5.

**Suitability:**

3

---

### Official Review · Reviewer_KjyG · 2024-06-04

**Rating:** 3
**Confidence:** 4

**Summary:**

This paper propose a reference-free image captioning metric, called HICEScore. HICEScore considers two aspects: global and local information.  For global matching, authors compute similarity between the image and text embedding. For local matching, they extract triplets by mobileSAMv2 and TextGraphParser for images and text respectively.

**Strengths:**

1. HICE includes a hierarchical score for image captioning evaluation: the global score for image-caption compatibility and the local score for region-phrase compatibility.

2. HICE can pinpoint incorrect textual mistakes and present unmentioned visual regions.

3. This paper is well-organized and well-written, making it easy to understand.

**Limitations:**

Dividing the evaluation of image and text matching into global and local levels is not innovative. InfoMetIC also uses the fine-grained information to evaluate image-caption pairs. From the results on open source benchmark, the effectiveness between HICE and InfoMetIC is very similar. One potential advantage of HICE is its ability to evaluate long captions. But Fig. 4 only shows the score of detailed captions generated by LLaVA-1.6 under multiple metrics, which does not demonstrate the advantages or disadvantages between HICE and InfoMetIC. Fig. 5 shows the correlation between human and HICE. This is not compared to InfoMetIC. I suggest that the author should compare more with the reference free method on detailed captions, and also publicly release the ratings of five human experts on LLaVA captions as a dataset.

**Suitability:**

3

---

### Meta-Review · Area_Chair_aDCR · 2024-07-03

**Recommendation:** Accept (Poster)
**Confidence:** 4

**Metareview:**

The paper initially received mixed ratings (2BA, BR, WR), which have then slightly moved towards positive after the rebuttal phase (3BA, 1WR). While the concerns raised from the negative reviewer are appropriate and sound, the paper still has value and is above the bar for an acceptance at ACM Multimedia. In particular, the usage of fine-grained semantic regions in an image captioning metric has never been proposed before, and the proposed metric beats the SoTA in terms of human correlation on different datasets. The number and kind of experiments and ablations performed is in-line with those done in recent image captioning metrics published in major venues (e.g. CVPR). Also, the quality of the paper and of the writing are above the bar. Nevertheless, I strongly encourage the authors to address the comment from all reviewers in the camera ready version of the paper. The paper, overall, can be accepted as poster.